# Examining the Moderating Effects of Work–Life Balance between Human Resource Practices and Intention to Stay

**Hsiao-Ping Chang [1,2], Chi-Ming Hsieh [3], Meei-Ying Lan [4] and Han-Shen Chen [1,2,*]**

[1] Department of Health Diet and Industry Management, Chung Shan Medical University, No. 110, Sec. 1, Jianguo N. Rd., Taichung City 40201, Taiwan

[2] Department of Medical Management, Chung Shan Medical University Hospital, No. 110, Sec. 1, Jianguo N. Rd., Taichung City 40201, Taiwan

[3] International Bachelor Program of Agribusiness, National Chung Hsing University, 145 Xingda Rd., South Dist., Taichung City 40227, Taiwan

[4] Department of Tourism and Leisure Management, Chung Chou University of Science and Technology, No. 6, Lane 2, Sec. 3, Shanjiao Rd., Yuanlin Township, Changhua County 510, Taiwan

[*] Correspondence: allen975@csmu.edu.tw; Tel.: +886-4-2473-0022 (ext. 12225)

**Abstract:** Offering services to clients by staff is a major feature of the hotel industry. Therefore, maintaining high-quality and stable services is critical for hotels to stay competitive. As for hotel human resource management, how to effectively increase employee retention is crucial because it not only enhances organizational performance but also reduces personnel cost. In this study, the researchers used structural equation modeling to explore the relationship between job embeddedness, organizational commitment, and intention to stay in tourist hotel interns. Furthermore, work–life balance was used as the moderating variable between organizational commitment and intention to stay. The study subjects were interns who had completed between half and one year of an internship at a tourist hotel and were going to graduate from the school upon completing the internship. The results indicated that job embeddedness has a significant and positive effect on organizational commitment and intention to stay whereas organizational commitment mediates the relationships between job embeddedness and intention to stay. In addition, work–life balance moderates interns' intention to stay. This study provides the hotel industry with useful management guidelines for retaining employees and improving competitiveness.

**Keywords:** talent management; job embeddedness; work–life balance; organizational commitment; strategic human resource management

## 1. Introduction

Attracting and retaining talent has always been the focus of human resource management (HRM) [1]. In particular, as the hotel industry is labor intensive, its high employee turnover rates have been a constant and difficult challenge for hotel human resource management because low retention rates increase organizational costs, thereby compromising organizations' competitive advantage [2]. For today's hospitality industry, how to enhance employees' organizational commitment and boost top employee retention is a major task. Due to staff shortages, internships have become a major source of labor for entry-level positions in the hospitality industry. Studies have shown that internships bridge students from school to the workplace by providing them with hands-on experience, improving their professional competence and increasing their employment opportunities. For hotels, internships reduce hotel labor costs and burden [3–5].

Hotel interns are often assigned to frontline service jobs, which involve frequent customer interaction and require them to manage various customer needs. Compared with other service industries, the hospitality industry has more intensive customer interaction [6,7]. Studies have shown that some students who have completed an off-campus internship or graduated decide to switch to another career path instead of staying in the hospitality industry [8–10]. Tourist hotels are serious about service quality and, therefore, they pay attention to their staff quality. Nevertheless, the high employee turnover rates in the hospitality industry can compromise hotel services by making service quality unstable, reducing hotel profitability, and decreasing customer satisfaction. Hinkin and Tracey asserted that each employee who quits will cause the organization to lose anywhere from thousands of US dollars to more than twice the annual salary of that employee [11]. In employee turnover studies, researchers have used the variables of work satisfaction and organizational commitment to explain employees' intention to leave and suggested that employees' intention to leave can be lowered by elevating their work satisfaction and organizational commitment [12–14]. Nonetheless, these studies have failed to explain why employees who are satisfied with their jobs are still tempted to switch jobs. By integrating employee resignation-related theories and numerous researchers' viewpoints, Mitchell, Holtom, Lee, Sablynski, and Erez proposed the concept of job embeddedness to explain this quitting behavior [15]. They further suggested that the various contacts established by employees in the organization increase the cost of their leaving; the greater the embeddedness of an employee, the lower the intention to leave. Studies have indicated that compared with variables such as organizational commitment and work satisfaction proposed by other researchers, job embeddedness can better explain employees' intention to leave [16,17]. Therefore, this study introduced job embeddedness to the hospitality industry to offer a more precise explanation of new employees' intention to stay; thus, hotels will be able develop effective strategies accordingly to boost their new employees' intention to stay and retain top talent.

Studies on job embeddedness have been more focused on validating the association between job embeddedness and intention to leave; few have examined the types of organizational behavior or scenarios facilitating the development of job embeddedness. Therefore, this study examined the association between job embeddedness and organizational commitment as well as that between organizational commitment and intention to stay from four aspects: job embeddedness, organizational commitment, work–life balance, and intention to stay. Furthermore, the researchers explored whether work–life balance moderates the effect of job embeddedness on intention to stay. Overall, this study aimed to provide human resource departments with useful information for retaining their top employees.

## 2. Literature Review

### 2.1. Job Embeddedness

Mitchell et al. suggested that job embeddedness is similar to an integrative network system where employees form indivisible links with people, things, and other networks of their organizations [15]. According to these researchers, higher density networks and stronger links will make employees more reluctant to change their jobs when they are offered a new one because they want to stay connected in the indivisible network of their organizations. In general, employees with higher job embeddedness are more willing to stay [18]. Mitchell et al. postulated that three key elements exist in job embeddedness [15]. The first is the content of the links between employees and other people or activities in their organizations or communities; the second is the level of similarity or fit of employees' lifestyles with their work and residential area; and the third is what employees must relinquish if they leave the job or community. In addition, Mitchell et al. asserted that three critical aspects exist for job embeddedness: links, fit, and sacrifice [15]. Links refers to the levels of interaction and connections—formal or casual—between individuals and their organizations or other members of the organizations. These links can be further divided into on-job links (which concern employees'

participation in group activities in the organization as well as other formal or casual interaction (e.g., business collaboration) with members of the organization) and off-job link (which are links formed when employees interact or exchange information with friends in their communities). Stronger links are suggested to equal higher levels of job embeddedness. Moreover, increasing job embeddedness will reduce the likelihood of people quitting their jobs. Compatibility refers to employees' perception of how compatible they are with or how fit they are for the organizations or environment of their organizations. On-job compatibility refers to how compatible a member's personal values, career goals, and plans for the future are with the knowledge and skills required for the organizational culture and work. High compatibility will make an employee less likely to leave the organization. External work compatibility refers to the level of fit to the residential area or community perceived by an individual, especially in terms of the culture, atmosphere, and leisure activities. The more suitable the activities or environment are for an individual, the higher the compatibility. Sacrifice refers to the benefits and values lost when an employee leaves the organization or community. In other words, it is the opportunity cost for quitting one's job. When an employee leaves a job, he or she may lose premium benefits, interpersonal interaction, and job promotion opportunities. In addition, the employee may have to change their living habits and lifestyle. Holtom and Inderrieden divided job embeddedness into on-job and off-job embeddedness [17]. On-job embeddedness concerns one's work organization and job-related matters, whereas off-job embeddedness concerns one's community and events outside of work. In this study, job embeddedness was examined from three aspects: organization fit, organizational sacrifice, and organizational links.

### 2.2. Organizational Commitment

Organizational commitment refers to psychological links between employees and their organizations. Robbins viewed organizational commitment as a work attitude reflecting the level at which employees identify themselves with their organizations and organizational goals, as well as the level at which they want to be a member of the organization [19]. Angle and Perry defined organizational commitment as an individual's level of care and loyalty for their organizations [20]. A person with a higher level of organizational commitment is more devoted to his or her organization. Reyes argued that organizational commitment concerns identifying with the values of one's organization, having a strong faith in that organization, and being inclined to viewing organizational commitment as a sense of belonging to and behavior toward the organization and their willingness to contribute to improve the organization's performance and effectiveness [21]. Shreya and Rajib asserted that organizational commitment concerns employees' willingness to work for their organizations, and when their organizations become less supportive, their organizational commitment will drop [22]. In addition, reducing employees' work stress can improve their organizational commitment. Robinson, Kralj, Solnet, Goh, and Callan defined organizational commitment as an individual's intention to stay in the organization [23].

### 2.3. Intention to Stay

According to Yoshimura, intention to stay concerns employees' willingness to and likelihood of staying in the organization by adopting a more positive attitude [24]. Intention to stay focuses not only on retaining an organization's critical talent, but also on eliciting positive work behavior in members of the organization. Reyes defined intention to stay as employees wishing to remain members of the organization or their willingness to stay in the organization [21]. Tett and Meyer explained intention to stay as employees' willingness to stay in the organization after cautious consideration [25]. Price, Rupp, Posey, Mueller, and Johnson suggested that intention to stay concerns employees' willingness to stay in the organization to work with their colleagues [26]. However, Coetzee and Stoltz argued that intention to stay concerns employees' loyalty to the work conditions and environment of their organizations as well as their willingness to remain working with colleagues in their organizations [27].

## 2.4. Relationships between Job Embeddedness, Organizational Commitment, and Intention to Stay

Lee et al. revealed that an individual may identify with their organization because of salary, status, job autonomy, or friendship with colleagues, and become reluctant to leave that organization [12]. This is called job embeddedness. Robinson et al. indicated that both organizational sacrifice in on-job embeddedness and community links in off-job embeddedness have significant effects on organizational commitment [23]. According to these studies, employees' organization fit has a positive effect on job embeddedness, and the greater the organization fit, the more positive this effect. Furthermore, these studies have indicated that job embeddedness mediates the positive relationships between organization fit and organizational commitment. Therefore, the present study proposed the following hypotheses:

**Hypothesis 1 (H1a).** *The greater the organization fit, the greater the organizational commitment.*

**Hypothesis 1 (H1b).** *The greater the organizational link, the greater the organizational commitment.*

**Hypothesis 1 (H1c).** *The greater the organizational sacrifice, the greater the organizational commitment.*

Griffeth, Hom, and Gaertner suggested that organizational commitment can effectively predict employees' job quitting behavior [28]. Low employee organizational commitment indicates more quitting behavior in employees. Furthermore, when employees identify with their organizations and its goals and wish to be members of their organizations, organizational commitment will be negatively associated with the rate of employee absenteeism [19]. Thus, organizational commitment is an emotional expression of belongingness, identification, and participation [29]. Members being highly committed to their organizations improves members' sense of solidarity, and the organizations' competitiveness will also improve. By contrast, if their commitment is low, they will feel insecure within their organizations and be more likely to quit. Perryer, Jordan, Firns, and Travaglione indicated that organizational commitment can effectively predict employees' intention to stay [30]. Porter, Steers, Mowday, and Boulian asserted that in comparison with job satisfaction, organizational commitment can better predict employees' intention to stay [31]. These studies suggested that organizational commitment is an antecedent variable of employees' intention to stay and, moreover, organizational commitment is positively associated with intention to stay. Thus, the higher the employees' organizational commitment, the greater their intention to stay. As a result, the present study proposed the following hypothesis:

**Hypothesis 2 (H2).** *The greater the organizational commitment, the greater the intention to stay.*

Studies have shown that job embeddedness has a high explanatory power for employee retention, and these two are positively correlated [15,32]. Therefore, higher levels of job embeddedness can more effectively reduce employees' job quitting behavior and make employees more willing to stay. Harari, Reaves, and Viswesvaran indicated that the organizational dimension of job embeddedness is a crucial indicator of employee performance [33]. Other studies have suggested that for the organizational dimension, an individual's intention to stay is affected by his or her job satisfaction and organizational commitment [34]. From a link perspective, employees' intention to stay can be improved by boosting their sense of commitment through enhancing their job content, work teams, and interaction with colleagues [15]. In terms of the level of fit, a poorer fit between employees and their organizations was indicated to be associated with higher job dissatisfaction and lower intentions to stay [35]. For organizational sacrifice, economic loss is the first challenge that people quitting their jobs must face. Therefore, the traditional attitude-based model has included and prioritized the factor of losing economic benefits and suggested that greater economic benefits are associated with higher job satisfaction and relatively higher intention to stay [36]. Some other types of organizational sacrifice and loss to be considered are job stability, job promotion opportunities, and interpersonal

relationships, which are all involved in employees' job-related decision-making. Therefore, the present study proposed the following hypotheses:

**Hypothesis 3 (H3a).** *The better the organization fit, the higher the intention to stay.*

**Hypothesis 3 (H3b).** *The higher the level of organizational link, the higher the intention to stay.*

**Hypothesis 3 (H3c).** *The greater the organizational sacrifice, the higher the intention to stay.*

Jiang, Liu, McKay, Lee, and Mitchell demonstrated that higher levels of job embeddedness are associated with higher organizational commitment [37]. Tang et al. demonstrated that a significant and positive association exists between job embeddedness and organizational commitment [38]. Kalidass and Bahron consistently showed a positive effect of organizational commitment on intention to stay [39]. Allen chose organizational socialization strategies, on-job embeddedness, and intention to leave as measures, and revealed that new employees' on-job embeddedness and intention to leave are negatively correlated [40]. Second, organizational socialization strategies can enhance new employees' job embeddedness. Moreover, organizational socialization strategies were found to be more strongly associated with on-job embeddedness than off-job embeddedness. Reiche, Kraimer, and Harzing considered that repatriated employees with higher levels of job embeddedness with their parent companies have lower intention to leave [41]. Allen and Shanock chose job embeddedness, perceived organizational support, organizational commitment, and voluntary redundancy as measures, and found that both perceived organizational support and job embeddedness affect new employees' organizational commitment and voluntary redundancy [42]. Robinson et al. showed that the organizational sacrifice in on-job embeddedness and the community links in off-job embeddedness have significant effects on organizational commitment and intention to leave [23]. More specifically, organizational sacrifice has a significant and negative effect on intention to leave, whereas community link has a significant and positive effect on intention to leave.

These findings suggested that job embeddedness affects employees' organizational commitment, and that the higher the employees' organizational commitment, the higher their intention to stay. Therefore, the present study proposed the following hypotheses.

**Hypothesis 4 (H4a).** *Organization fit increases intention to stay through organizational commitment.*

**Hypothesis 4 (H4b).** *Organizational link increases intention to stay through organizational commitment.*

**Hypothesis 4 (H4c).** *Organizational sacrifice increases intention to stay through organizational commitment.*

*2.5. Moderating Effect of Work–Life Balance*

In recent years, work–life balance has been found to be very important in attracting and retaining talent [43,44]. Work–life balance means an employee is achieving balance between work, home, and other life roles [45,46]. Work–life balance is defined as the accomplishment of satisfactory experiences in all domains of life. Attaining satisfying experiences in all domains of life requires an equal distribution of personal resources such as energy, time, and commitment across all domains [47]. If the responsibilities, pressures, and obligations of employees at work are too high, or the working hours are too long and the resources are not evenly distributed, they may oppress the quality of life and lead to poor physical and mental state. Therefore, work–life balance is the employee mentality that must be emphasized in organizational management [48]. Greenhaus and Beutell [49] have pointed out that work–life imbalances can cause problems such as reduced employee productivity, low morale, lateness, absence, etc., and may lead to employee turnover. If employees can maintain a good balance between work and personal life, it will be beneficial to both the company and the employees themselves in

the long run. According to Hayman's [50] study, work–life balance is defined as the psychological satisfaction of individuals, and it can comprehensively improve self-efficacy. Work–life balance is negatively correlated with the number of overtime work and the working hours of the individual. The longer the number of overtime work and the number of working hours, the more unbalanced work and life will affect the employee's willingness to stay and the productivity of the work [51]. Organizational commitment and growth are pivotal for an individual from the point of view of career growth. They are obligatory for an individual to accomplish societal commitments, social responsibilities, and share time for the wellbeing of society. However, commitment towards self-development and sound health are essential for leading a peaceful life. Any mismanagement among personal, societal, and organizational commitments can end up with serious consequences in each of those areas. While definitions and explanations differ, work–life balance can be generally associated with equilibrium, or maintenance of an overall sense of harmony in life [52,53]. Malone and Issa refer to a person's level of organizational commitment as a reliable predictor of employee turnover, and work–life balance had a decisive impact on an employee's overall job satisfaction, organizational commitment, and willingness to stay [54]. Poor work–life balance has negative consequences on employees' health and wellbeing, as well as organizations' performance [55,56]. Hence, an imbalance between work and personal life causes higher stress that might also lead to greater turnover intention among employees [57,58]. As such, employees' ability to achieve work–life balance with organizational support should lead to higher job engagement, greater commitment, better job performance, and lower turnover rate [57,59]. According to these findings, work–life balance is associated with organizational commitment, and it affects intention to stay. In addition, the positive effect of organizational commitment on intention to stay is mediated by organizational commitment reinforcing work–life balance. Therefore, this study suspects that work–life balance has a moderating effect on the association between organizational commitment and intention to stay, and proposed the following hypothesis:

**Hypothesis 5 (H5a).** *Work–life balance reinforces the association between organizational commitment and intention to stay.*

## 3. Methodology

### 3.1. Sample and Data Collection

Convenience sampling was used to survey interns with at least 6 months of work experience in Taiwanese tourist hotels. We distributed 400 survey forms and 396 were returned. After eliminating 171 invalid survey forms, 225 valid survey forms remained for a valid return rate of 56%. SPSS 18.0 was used to perform descriptive statistical and reliability analyses, and AMOS 7.0 was used to conduct confirmatory factor analysis and structural equation modeling. Among the interns completing the survey, 91 were men (40.4%) and 134 were women (59.6%). In terms of academic department, most were from hospitality management ($N$ = 110, 48.8%), followed by food service management ($N$ = 50, 22.2%). For most interns completing the survey, the duration of their internships was 1 year ($N$ = 143, 63.5%), though 82 of the interns had half-year internships (36.4%). Most had a monthly salary of NT$20,001–25,000 ($N$ = 89, 39.5%), followed by NT$15,001–20,000 ($N$ = 78, 34.6%).

### 3.2. Measurement

Based on the literature review, Figure 1 presents the research framework for investigating the relationships between job embeddedness (organization fit, organization link, and organization sacrifice), organizational commitment, intention to stay, and work–life balance for interns in Taiwanese tourist hotels. Within this framework, the independent variables are organization fit, organization link, and organization sacrifice; intention to stay is a dependent variable; organizational commitment is a mediator variable; and work–life balance is a moderator variable. The 10-item scale in the organization

fit section of the questionnaire included items such as "The present job enables me to fulfill myself"; the 3-item scale in the organization link section of this survey included items such as "My colleagues and I have positive interaction at work"); and the 7-item scale in the organization sacrifice section included items such as "I am willing to improve my competence for this hotel". These were adapted from Mitchell et al. and Holtom and Inderrieden [15,17]; in these studies, the Cronbach's α for the organization fit, organization link, and organization sacrifice scales ranged from 0.85 to 0.89. The 6-item scale in the organizational commitment section of the survey included items such as "In the future, I am willing to expand my knowledge and enhance my skills to improve my work performance," and was adapted from Robinson et al. with a Cronbach's α of 0.85 [23]. The 5-item scale in the intention to stay section included items such as "I am willing to stay and work for this hotel," and was adapted from Coetzee and Stoltz with a Cronbach's α of 0.95 [27]. The 11-item scale in the work–life balance section included items such as "Work causes loss of personal life," and was adapted from Hayman with a Cronbach's α of 0.97 [50]. The Cronbach's α coefficients of these constructs were above the 0.70 level suggested by Nunnally, and in each scale, measurements were based on a 7-point Likert scale ranging from 1 (strongly disagree) to 7 (strongly agree) [60].

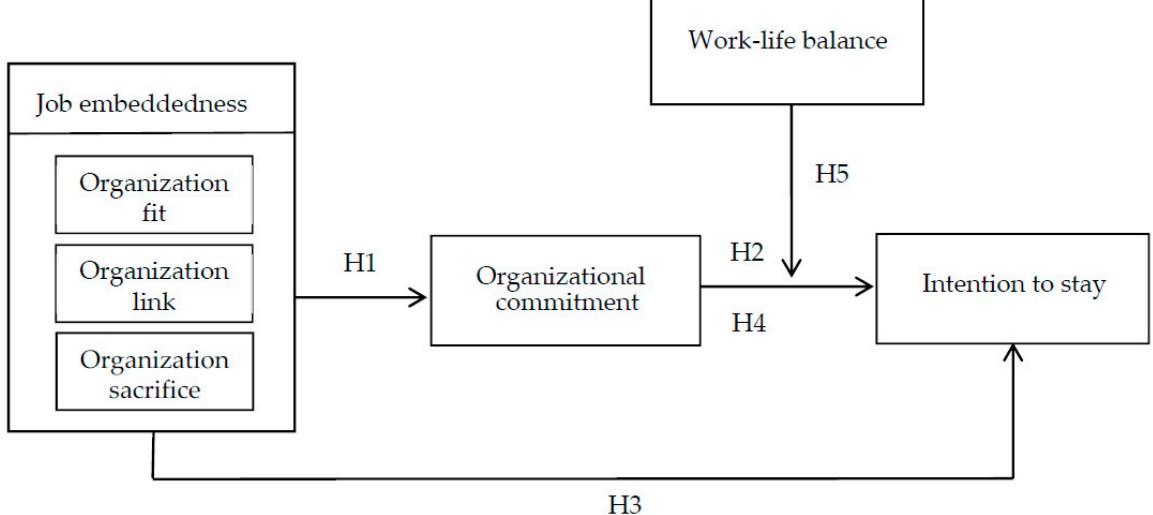

**Figure 1.** Research framework.

## 4. Results

### 4.1. Descriptive Analyses

Table 1 presents the descriptive statistics resulting from our analysis, which included means, standard deviations, and a correlation matrix. Consistent with our hypotheses, the correlations among organization fit, organization link, organization sacrifice, organizational commitment, work–life balance, and intention to stay were all significant. The Cronbach's α of each dimension was greater than 0.80, indicating good reliability [60]. Fornell and Larcker suggested that the composite reliability (CR) of latent variables should be greater than 0.60 [61]; a high CR of latent variables for an examined variable indicates that the examined variable is valid for use in measuring the latent variable. The CR of the variables in this study ranged from 0.69 to 0.93, indicating that this model had good internal consistency [62]. Fornell and Larcker and Bagozzi and Yi suggested that the average variance extracted (AVE) of potential variables should be greater than 0.50 [61,63]. Moreover, if the AVE of three to four out of five potential variables reaches 0.50, whereas the AVE of the remaining variables is greater than 0.30 or 0.40, this is acceptable. The AVE for each factor was between 0.43 and 0.76, which is higher than the benchmark of 0.3 recommended by Fornell and Larcker [61].

**Table 1.** The means, standard deviations, Cronbach's α, CR, AVE, and correlations of the examined variables.

| Variables | M | SD | CR | AVE | 1 | 2 | 3 | 4 | 5 | 6 |
|---|---|---|---|---|---|---|---|---|---|---|
| 1. Organization fit | 5.118 | 0.808 | 0.91 | 0.54 | **(0.890)** | | | | | |
| 2. Organization link | 5.690 | 1.212 | 0.69 | 0.43 | 0.259** | **(0.847)** | | | | |
| 3. Organization sacrifice | 5.026 | 1.005 | 0.90 | 0.63 | 0.723** | 0.259** | **(0.864)** | | | |
| 4. Organizational commitment | 5.451 | 0.855 | 0.93 | 0.71 | 0.464** | 0.141** | 0.464** | **(0.853)** | | |
| 5. Work–life balance | 5.691 | 0.856 | 0.90 | 0.53 | 0.348** | 0.385** | 0.347** | 0.417** | **(0.967)** | |
| 6. Intention to stay | 4.459 | 1.585 | 0.93 | 0.76 | 0.566** | 0.226** | 0.556** | 0.706** | 0.492** | **(0.945)** |

$N = 225$; ** $p < 0.01$; CR = composite reliability; AVE = average variance extracted; bold numbers in parentheses (diagonally) refer to the reliability of the variables (Cronbach's α).

### 4.2. Structural Equation Modeling and Empirical Analysis

We applied SEM using AMOS 18.0 to assess the path relationships among the examined constructs [64]. The results indicated that the measurement model provided a good fit for the data ($\chi^2$/df = 3.96, GFI = 0.912, AGFI = 0.917, CFI = 0.920, RMR = 0.031, RMSEA = 0.072). The GFI, AGFI, and CFI exceeded the recommended threshold of 0.90, and the RMSEA and SRMR values were below the cutoff value of 0.08 [65]. This indicated that the approach used in this study for modeling the examined data was appropriate. The hypotheses testing results from the model data are provided in Figure 2 and Table 2.

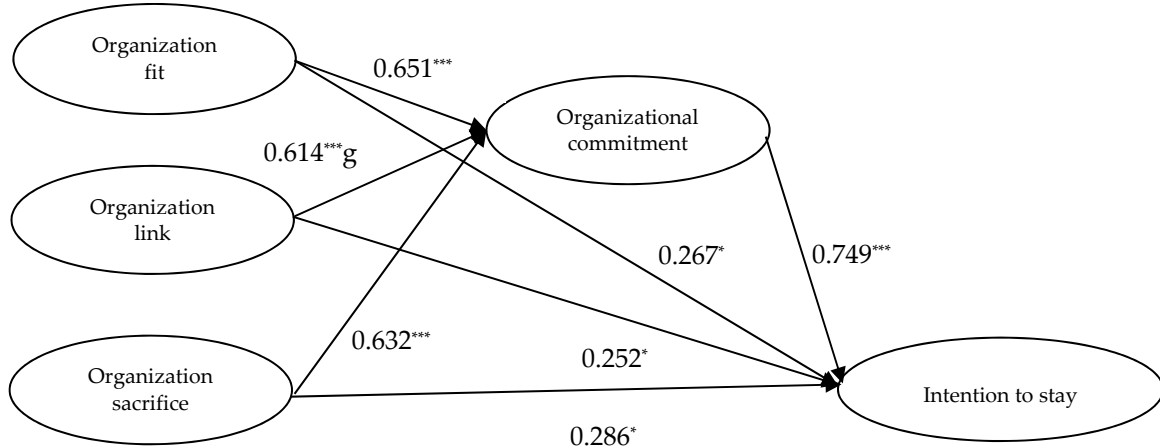

**Figure 2.** Results of the hypothesized model. Note. * $p < 0.05$; *** $p < 0.001$.

Hypotheses H1a, H1b, and H1c were supported. The results indicated that organization fit, organization links, and organization sacrifice of tourist hotel interns are significantly and positively associated with organizational commitment. For pathway coefficients, γ11 is 0.651 ($p < 0.001$), γ12 is 0.614 ($p < 0.001$) and γ13 is 0.632 ($p < 0.001$), respectively. These findings implied that enhancing interns' behavior as related to organization fit, organization links, and organization sacrifice is critical for tourist hotels to elicit organizational commitment in their interns. As for the levels of reinforcement, organization fit should be the highest, followed by organization sacrifice and then organization links.

Hypothesis H2 was supported. This meant that tourism hotel interns' organizational commitment and intention to stay are significantly and positively associated. The pathway coefficient β21 was 0.749 ($p < 0.001$), suggesting that enhancing interns' organizational commitment behavior will positively reinforce their intention to stay.

Hypotheses H3a, H3b, and H3c were supported. This meant that tourism hotel interns' organization fit, organization links, and organization sacrifice are significantly and positively associated with their intention to stay. For pathway coefficients, γ21 was 0.267 ($p < 0.05$), γ22 was 0.252 ($p < 0.05$), and γ23 was 0.286 ($p < 0.05$). This implied that tourist hotels should enhance interns' organization fit, organization links, and organization sacrifice behavior to positively reinforce their intention to

stay. In terms of levels of reinforcement, organization sacrifice should be the highest, followed by organization fit and then organization links.

These results revealed that a significant and positive association exists between job embeddedness and intention to stay. Therefore, this study further investigated whether organizational commitment could alter the effect of job embeddedness on intention to stay. Table 2 shows that the direct effect of organization fit on intention to stay was 0.267. With organizational commitment, the indirect effect between organization fit and intention to stay ($\gamma11 \times \beta21$) became 0488 ($p < 0.001$). Since the indirect effect was greater than the direct effect, a mediating effect occurred, and hypothesis H4a was supported. The direct effect between organization links and intention to stay was 0.252. With organizational commitment, the indirect effect between the two ($\gamma12 \times \beta21$) became 0460 ($p < 0.001$). Since the indirect effect was greater than the direct effect, a mediating effect occurred. Therefore, hypothesis H4b was supported. The direct effect between organization sacrifice and intention to stay was 0.286. With organizational commitment, the indirect effect between the two ($\gamma13 \times \beta21$) became 0.473 ($p < 0.001$). The indirect effect was greater than the direct effect and, therefore, a mediating effect occurred. Thus, hypothesis H4c was supported.

**Table 2.** The model's standardized regression weights, *t*-values, and hypothesis.

| Path | Standardized Regression Weight | *t*-Value | Hypothesis |
|---|---|---|---|
| Directed effect of the integrative model | | | |
| Organization fit → Organizational Commitment ($\gamma_{11}$) | 0.651 | 5.784*** | H1a* |
| Organization link → Organizational Commitment ($\gamma_{12}$) | 0.614 | 10.838*** | H1b* |
| Organization sacrifice → Organizational Commitment ($\gamma_{13}$) | 0.632 | 9.400*** | H1c* |
| Organizational Commitment → Intention to stay ($\beta_{21}$) | 0.749 | 18.050*** | H2* |
| Organization fit → Intention to stay ($\gamma_{21}$) | 0.267 | 2.008* | H3a* |
| Organization link → Intention to stay ($\gamma_{22}$) | 0.252 | 2.006* | H3b* |
| Organization sacrifice → Intention to stay ($\gamma_{23}$) | 0.286 | 2.552* | H3c* |
| Mediating effect of the integrative model | | | |
| Organization fit → Organizational Commitment → Intention to stay ($\gamma_{11} \times \beta_{21}$) | 0.488 | – | H4a* |
| Organization link → Organizational Commitment → Intention to stay ($\gamma_{12} \times \beta_{21}$) | 0.460 | – | H4b* |
| Organization sacrifice → Organizational Commitment → Intention to stay ($\gamma_{13} \times \beta_{21}$) | 0.473 | – | H4c* |
| $\chi^2$/df = 3.96, GFI = 0.912, AGFI = 0.917, CFI = 0.920, RMR = 0.031, RMSEA = 0.072 | | | |

$t > 1.96$, * $p < 0.05$; $t > 0.258$, ** $p < 0.01$; $t > 3.29$, *** $p < 0.001$; * indicates the hypothesis was supported.

### 4.3. Testing Moderating Effects

Hierarchical regression was used to analyze the moderating effect of work–life balance on the relationship between organizational commitment and intention to stay. The analysis results are presented in Table 3. The regression model M3 in this table shows that including the product of organizational commitment and work–life balance increased the coefficient of determination $R^2$ to 0.272, resulting in a difference ($\triangle R^2$) of 0.020. Moreover, the regression coefficient of organizational commitment × work–life balance reached significance ($\beta = 0.645$, $p < 0.001$), suggesting that work–life balance has a moderating effect on the relationship between organizational commitment and intention to stay. Subsequently, the researchers used the mean of work–life balance as a cutoff point to divide the sample in two: good work–life balance and bad work–life balance. This study analyzed each part separately and drew the regression model segment for organizational commitment and intention to stay (Figure 3). When interns perceived the work–life balance as good, the effect of organizational commitment and intention to stay was enhanced. Furthermore, when work–life balance was perceived as bad by interns, this enhanced the effect of organizational commitment and intention to stay. However, when comparing these two lines in terms of slope, work–life balance perceived as good by interns

could more accurately predict the effect of organizational commitment and intention to stay than the work–life balance perceived by interns as bad.

**Table 3.** Results of hierarchical regression analysis.

| Variables | Intention to Stay | | |
|---|---|---|---|
| | **Model 1** | **Model 2** | **Model 3** |
| Step 1: Independent variable | | | |
| Organizational commitment | 0.749*** | 0.257 | 0.137 |
| Step 2: Moderator | | | |
| Work–life balance | | 0.630*** | 0.366*** |
| Step 3: Interaction | | | |
| Organizational Commitment × work–life balance | | | 0.645 *** |
| $R^2$ | 0.174 | 0.252 | 0.272 |
| $\triangle R^2$ | | 0.078 | 0.020 |
| F | 47.004 *** | 37.426 *** | 26.121 *** |

\* $p < 0.05$; \*\* $p < 0.01$; \*\*\* $p < 0.001$.

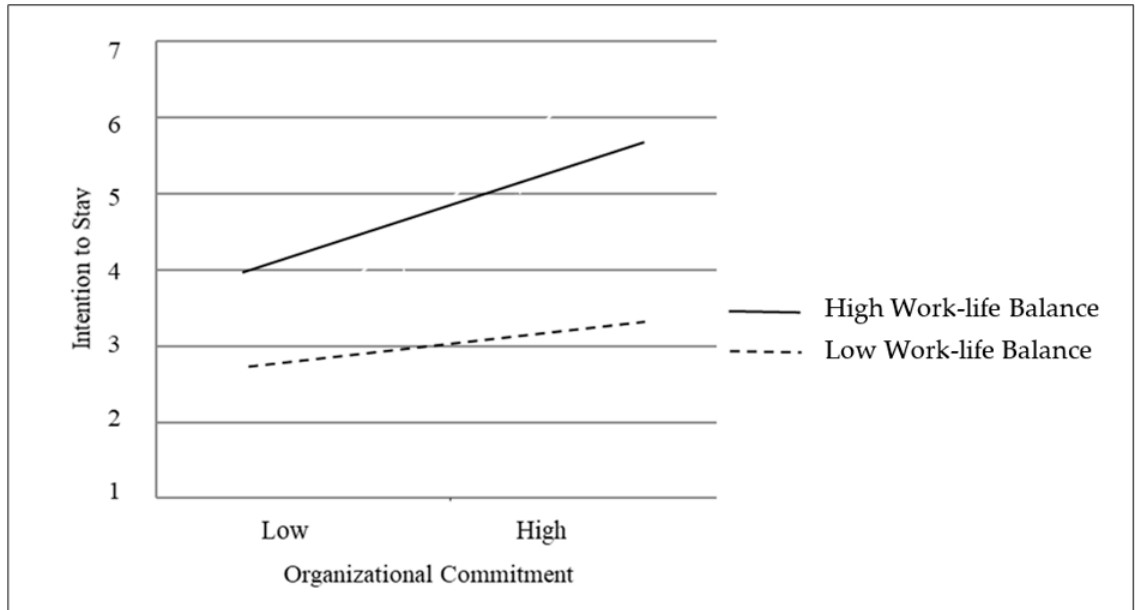

**Figure 3.** The interactive effect of organizational commitment, intention to stay and work–life balance.

## 5. Discussion

A significant and positive association existed between job embeddedness (organization fit, organization links, and organization sacrifice) and organizational commitment. This finding is consistent with Robinson et al. [23]. For managerial implications, this finding suggests that job embeddedness as perceived by interns can affect their organizational commitment. Therefore, when the goals, welfare, vision, and work environment of a hotel are similar to the needs of the employees, a higher level of organizational commitment to the hotel will be elicited from interns. Second, a significant and positive association existed between organizational commitment and intention to stay, and the finding here is consistent with Gieter, Hofmans, and Pepermans [66] and Hansel, Froese, and Pak [67]. For managerial implications, this finding suggests that interns with higher organizational commitment are more willing to stay and work for their hotels. Therefore, hotels should implement comprehensive education and training and value the development of their interns. Moreover, setting

up a complete rotation system, enhancing organizational solidarity among interns, and forming positive brand image are crucial for boosting interns' organizational commitment and enhancing their willingness to stay. Third, a significant and positive association existed between job embeddedness and intention to stay; studies such as those by Robinson et al. and Tang et al. had similar findings [23,38]. For managerial implications, this result indicated that the level of job embeddedness perceived by interns affects their intention to stay, and when the goals, welfare, vision, and work environment of a hotel are closer to the needs of the employees, interns will be more willing to stay and work for the hotel after completing the internship. Fourth, organizational commitment has a partial mediating effect on the association between job embeddedness and intention to stay; Ramesh and Gelfand showed a similar viewpoint: although job embeddedness is directly associated with intention to stay, enhancing organizational commitment is more effective at increasing interns' intention to stay [68]. For managerial implications, this finding indicated that when hotels implement effective policies that satisfy interns by increasing their identification with the organization, interns with a higher organizational commitment will be more willing to stay and work for the hotel. Fifth, work–life balance has a moderating effect on the association between organizational commitment and intention to stay. Consistent with Chang, Liou, and Yang [69] and Noor [70], this study indicated that improving work–life balance will enhance employees' satisfaction with their self-efficacy and service quality and this improvement, in turn, will enhance employees' organizational commitment, making them more willing to stay. For managerial implications, this finding indicated that hotels should offer a good work–life balance for interns to identify with their hotels and would likely improve productivity and job satisfaction. This will help interns accept their hotel's various policies, thereby boosting interns' organizational commitment and intention to stay.

## 6. Conclusions

### 6.1. Findings

This study demonstrated that organizational commitment has the most potent effect on intention to stay and, therefore, a core task for hotels is to enhance interns' organizational commitment to increase their intention to stay. This study presented some distinct viewpoints and suggestions: (1) Provide interns with room for development: Similar to regular employees, interns care about whether their ideas are accepted by their hotels. Therefore, hotels should help interns unleash their talent and be aware of their need for competency development. For interns with superior performance to their peers, hotels should allow them to learn high-level work and assure them that their salary will be higher if they stay. Moreover, encouraging and assisting students with suboptimal performance is crucial to boost their trust in the organization as well as their sense of achievement; thus, they will perceive that the organization values them and is devoted to their relationship. Enhancing interns' sense of achievement will make them more willing to contribute, even if their internships are as short as half to 1 year, as well as to stay after completing their internships. (2) Provide routine education and training: Since the duration of internships tends to be short, related education and training are mostly arranged before, and rarely during, the internship. Therefore, a good idea is for hotels to offer routine education and training to provide interns with opportunities to expand their professional knowledge and increase their skills which, in turn, can improve their work performance. The objective is to enhance interns' sense of belonging with their organizations and reinforce related behavior, improve organizational effectiveness and efficiency, and encourage interns to make more contributions to their organizations. Interns' organizational commitment will be strengthened when they show a more involved attitude with their organizations. Moreover, these interns are more likely to accept and identify themselves with organizational objectives and values and become more willing to stay and work for the organization. (3) Establish a positive brand image and enhance interns' sense of solidarity with the organization: Hotels establishing a positive brand image is critical because a it will not only attract more customers, but also convince their interns that staying to work for the hotel is good for their

future career. Moreover, management should actively enhance interns' sense of solidarity by caring for their interpersonal relationships occasionally, helping them adapt to the culture of the organization, and promoting a more positive operation of the union of interns. These measures facilitate establishing bilateral communication between the hotel and its interns. Another suggestion is to routinely hold activities with other organizations and encourage intern participation. This type of activity increases interns' affection for their organization and wins their trust. Interns will learn that their organizations are friendly, caring, and positive. These activities will help foster a sense of belonging, making interns feel that they are part of the family. Consequently, they will be more willing to stay because they know that their needs can be better fulfilled, and that they are valued by their organizations. (4) Establish a positive work–life balance: This study found that work–life balance had a moderating effect on the association between organizational commitment and intention to stay. Therefore, hotels' human resource department should assist employees in developing a good work–life balance. Moreover, helping those interns develop more mature and comprehensive thinking is crucial for expressing their talent and creating a positive atmosphere for each and every employee, thereby enabling them to develop optimal performance and win recognition. A key is for hotels to understand the level of their interns and detect through organizational operations whether interns' perception of the organization is consistent with the expectations of the organization. If not, then hotels should work on improving themselves and resolving issues reflected by interns to improve the working environment for interns and enhance their intention to stay.

### 6.2. Limitations and Further Research

This study had some limitations. Its subjects were tourist hotel interns who had completed an internship of at least half a year. Sampling was difficult because of the associated time and costs. The small sample size and the use of convenient sampling may have limited the scope of sampling, preventing the study results being applied to the overall tourist hotel industry. Second, whether interns are willing to stay or not is subjectively determined by interns through a complex mental process. This study used a questionnaire survey approach to probe this mental aspect of interns' decision-making; however, this approach may not have been sophisticated enough to comprehensively and accurately assess their true feelings toward work. Third, this study adopts cross-sectional analysis rather than longitudinal analysis, suggesting that future research can perform longitudinal analysis, using different time points or periods to facilitate causality determination. Lastly, in this study's analyses, the focus was on the effects of on-job embeddedness on employees' organizational commitment and intention to stay. Considering that numerous internal and external factors are known to affect organizational commitment and intention to stay—other than on-job embeddedness and work–life balance—subsequent studies should select and examine other internal or external factors to widen the scope of understanding in these areas.

**Author Contributions:** Four co-authors together contributed to the completion of this article. H.-P.C. was the first author, who analyzed the data and drafted the manuscript; C.-M.H. contributed to reviewing the manuscript and revising the results and conclusion; M.-Y.L. contributed to reviewing and revising the literature, results, and conclusion; and H.-S.C. acted as corresponding author on their behalf throughout the revision and submission process.

**Funding:** This research received no external funding.

**Conflicts of Interest:** The authors declare no conflict of interest.

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
