# Peer review of "Examining the Moderating Effects of Work–Life Balance between Human Resource Practices and Intention to Stay"

_sustainability, doi:10.3390/su11174585_

Round 1

Reviewer 1 Report

The research is interesting and the results are useful to demonstrate that policies companies are very important ti support employee's well-being but also the "productivity" of the company itself.

Author Response

Thank you for your encouragement, we will continue to work hard.

Reviewer 2 Report

Abstract states "to explore causality" - this is only possible if you use longitudinal data. And you dont, so rewrite this!

Also, "A total of 300 questionnaires was distributed, and the valid response rate was 75%." - Just report the number of respondents - details around response rate etc. can go in the methods section (but not abstract).

Page 1 states "Zopiatis (2007) suggested that" - needs the citation number (of the reference) after the author

The introduction is strong and lays a persuasive argument for the paper. Well done.

Page 4 states "Morris and Sherman (1981) suggested that organizational commitment can effectively predict employees’ job quitting behavior [28]."

This reference is 38 years old! The paper does seem to be missing some references to meta-analyses in the turnover field, like 

Griffeth, R. W., Hom, P. W., & Gaertner, S. (2000). A meta-analysis of antecedents and correlates of employee turnover: Update, moderator tests, and research implications for the next millennium. Journal of Management, 26(3), 463-488.  

I'd suggest this reference could help shape the org commitment argument! It might also be used for arguing for your control variables.

Hypotheses 4 (a-c) need work. Is this mediation? The shaping of the argument is more like moderation! You want to say that while job embeddedness is linked to employee retention, as is org commitment, it is expected that org commitent will mediate the influence of the job embeddedness dimensions. something like that!

Page 5 states "Work-life balance is also act of corresponding between individual’s three aspects (personal life, societal life and organizational life) of life which is described [48]."

The end part of this sentence doesn't make sense. "which is described..." where? what?! unclear!

The moderation argument needs strengthening. Are there any theoretical links that support his?

Note reference 46 is from the journal Research and Practice in Human Resource Management - so include it ALL in italics [you only do the last 3 words!]

The analysis is fine and results look good. Please report the r2 value for intentions to stay. Must say the org commitment- intentions to stay are VERY highly correlated at .70! 

It is also unclear why you didn't enter the interaction into the SEM model and run it from there? See:

Haar, J. M., Russo, M., Sune, A., & Ollier-Malaterre, A. (2014). Outcomes of work-life balance on job satisfaction, life satisfaction and mental health: A study across seven cultures. Journal of Vocational Behavior, 85(3), 361-373.

I'm also surprised that in that analysis, work-life balance mediates the effect of organizational commitment! That seems VERY strange and not as expected. I suggest doing that analysis in SEM might derive more accurate effects!

Overall, the discussion is solid.

Need to mention the cross-sectional nature of the data in Limitations and how this means causality cannot be determined.

Finally, it is hard to interpret the work-life balance interaction given that analysis is outside of SEM, and thus ignored embeddedness. I also am wary that org commitment direct effect was fully mediated - that seems VERY unusual.
